# Overview of Physicochemical Properties of Nanoparticles as Drug Carriers for Targeted Cancer Therapy

**DOI:** 10.3390/jfb13040196

**Published:** 2022-10-20

**Authors:** Vugar Yagublu, Aynura Karimova, Javahir Hajibabazadeh, Christoph Reissfelder, Mustafa Muradov, Stefano Bellucci, Adil Allahverdiyev

**Affiliations:** 1Department of Surgery, Medical Faculty Mannheim, University of Heidelberg, 68167 Mannheim, Germany; 2Nanoresearch Laboratory, Baku State University, AZ 1148 Baku, Azerbaijan; 3Istituto Nazionale di Fisica Nucleare—Laboratori Nazionali di Frascati, Via E. Fermi 54, 00044 Frascati, Italy; 4Vali Akhundov National Scientific Research Medical Prophylactic Institute, AZ 1065 Baku, Azerbaijan

**Keywords:** nanoparticles, tumor microenvironment, drug carriers, protein corona, surface modification

## Abstract

The advent of nanotechnology has brought about revolutionary innovations in biological research techniques and medical practice. In recent years, various “smart” nanocarriers have been introduced to deliver therapeutic agents specifically to the tumor tissue in a controlled manner, thereby minimizing their side effects and reducing both dosage and dosage frequency. A large number of nanoparticles have demonstrated initial success in preclinical evaluation but modest therapeutic benefits in the clinical setting, partly due to insufficient delivery to the tumor site and penetration in tumor tissue. Therefore, a precise understanding of the relationships betweenthe physicochemical properties of nanoparticles and their interaction with the surrounding microenvironment in the body is extremely important for achieving higher concentrations and better functionality in tumor tissues. This knowledge would help to effectively combine multiple advantageous functions in one nanoparticle. The main focus of the discussion in this review, therefore, will relate to the main physicochemical properties of nanoparticles while interacting within the body and their tuning potential for increased performance.

## 1. Introduction

The broad application of nanoparticles in biological research techniques and medical practice has brought about revolutionary innovations and played a key role in resolving many problems in these realms [1,2,3]. All applications of nanoparticles in biomedicine can generally be divided into two major groups: therapeutic and diagnostic. Therapeutic approaches include the use of nanoparticles to ensure the targeted delivery of drugs or genetic materials to the target tissues. For instance, magnetic nanoparticles with remarkable superparamagnetic properties could be controlled via an applied external gradient of the magnetic field [4,5], and this “action at a distance” facilitates the transport and delivery of drugs to the corresponding tumor sites. Moreover, nanoparticles can ensure the delivery of drug molecules even to subcellular structures, such as cell nuclei or mitochondria [6]. These capacities distinguish them from others, based on factors including reduced off-target drug toxicity and the need for the administration of repetitive doses. Furthermore, magnetic nanoparticles are used for other treatment applications, such as magnetic hyperthermia [7]. It is considered one of the promising cancer treatment tools, where heating, as a result of various fluctuations of the particle magnetic moments, causes the death of tumor cells [8].

Nanoparticles are also considered a prime option for the immunotherapy of various tumors due to their multisided physicochemical and structural characteristics [9,10]. Nanomedicines—particularly target immune cells or immune reactions, such as liposomes, polylactic-co-glycolic acid (PLGA) nanoparticles, dendrimers, gold and carbon nanoparticles, micelles, polymeric nanoparticles, quantum dots, and microneedles—have been reported [10].

Certain types of nanoparticles allow for the imaging of individual, pathologically altered cells and even molecules that are markers of disease. It has already been reported that minor metal oxide nanoparticles, such as iron oxide (Fe_3_O_4_ [11]), gadolinium oxide (Gd_2_O_3_ [12,13]), and manganese oxide (Mn_3_O_4_ [14])—due to their remarkable physical properties—could be applied as contrast agents for more accurate imaging. Moreover, the fluorescence emission properties of nanoparticles, which are dependent on their sizes, make them indispensable for molecular imaging; this provides opportunities for detecting diseases as early as possible at the molecular level [15,16].

Unfortunately, despite all these advantages, nanoparticles used for therapeutic purposes cannot overcome the various biological barriers that exist on the way from their entry into the blood circulation to the target sites, and this reduces their effectiveness in clinical settings.The barriers that prevent intravenously administered nanoparticles from reaching their target are shown schematically in Figure 1.

These include the reticuloendothelial system (RES), comprised of the liver, spleen, lungs, and bone marrow [17,18]; the endothelium of blood vessels within the target tissues [19]; and the extracellular matrix (ECM) of the target tissue [20,21], which is a highly organized and complex network of structural proteins, proteoglycans, and glycosaminoglycans. Recent studies have demonstrated that nanoparticles can reach a tumor passively through the leaky vasculature surrounding the tumor because of the enhanced permeability and retention (EPR) effect, which opens new perspectives for overcoming tumor chemoresistance [22]. Furthermore, the physicochemical properties of nanoparticles play decisive roles in overcoming all three barriers, which we discuss below. In addition, we consider the dynamics of protein corona formation and its impact on nanoparticle behavior in a biological environment.

## 2. Importance of Physicochemical Properties of Nanoparticles

A large number of nanoparticles have demonstrated initial success in preclinical evaluation, but modest therapeutic benefit in clinical settings, partly due to insufficient delivery to the tumor site and penetration of tumor tissue. Therefore, a precise understanding of the relationships betweenthe physicochemical properties of nanoparticles and their interaction with the surrounding microenvironment in the body is extremely important for achieving higher concentrations and better functionality in tumor tissues.

Nanoparticles are characterized by unique properties, such as small spatial dimensions, the possibility of specific surface binding sites for various ligands, and the manifestation of resonance-related absorption under different types of external energy influences, with subsequent relaxation processes [23]. These features provide a multifunctional platform for the development of various biomedical applications.

The main powerful feature of nanoparticles is demonstrating fundamentally different physicochemical properties than their micro- and macroscopic analogs—ones that previously have not been seen in certain contexts, for instance, in the human body. The changes after exposure to the biological fluids could be both advantageous and highly deleterious for the expected functionality of nanoparticles in the target site, which should be taken into consideration during the design process. Synthetic nanoparticles can penetrate “the walls” inside cell systems—organelles or nuclei—and interact with natural nano-sized targets in unpredictable ways due to their tunable physicochemical characteristics. The most important characteristics influencing the interaction of nanoparticles with biological systems include their size, shape, hydrophilicity/hydrophobicity properties, surface coating composition, the presence of functional groups on them, and their charge (see Figure 2).

### 2.1. Size of Nanoparticles

The small size of nanoparticles is one of the main features allowing them to pass through biological barriers;easily enter cells; and translocate across cells, tissues, and organs [24,25]. This significantly influences their therapeutic and diagnostic value [26]. Certain factors, such as cellular uptake [27] and elimination [28], intracellular trafficking [29], cytotoxicity [30,31], tumor penetration [32], and blood circulation half-time [33], are influenced by nanoparticle size.

Nanoparticles, as polar molecules, enter the cells by employing an endocytotic pathway, whereas the cell membrane is mostly permeable for small non-polar molecules [34]. It is generally believed that smaller nanoparticles are taken up by the cells faster than the larger ones [25]; however, as the particle size decreases, the surface area to volume ratio increases [35]. This contributes to an elevation in both the rate and extent of cellular uptake but promotes rapid elimination through the kidneys at the same time [36,37,38].

Smaller nanoparticles have exhibited higher toxicity than larger ones. Silver nanoparticles were synthesized into three different sizes (~10 nm, 50 nm, and 100 nm) and were studied in several cell lines, including an MC3T3-E1 clonal murine cell line of immature osteoblasts derived from mice and PC12 rat pheochromocytoma cells.Nanoparticles with a size of 10 nm were found to cause more cell apoptosis than larger nanoparticles. The other group, using the BEAS-2B normal bronchial epithelial cell line and silver nanoparticles (AgNPs) of 10, 40, 50, and 75 nm sizes, showed size-dependent cytotoxicity, where only the 10 nm particles affected the cell viability of human lung cells [39].

Other in vitro studies have shown maximum cellular uptake in nonphagocytic cells, within the size range of 10–60 nm regardless of nanoparticle core composition or surface charge [33]. Nanoparticles with sizesdeviatingfrom this range could demonstrate undesired effects. For instance, some nanoparticles with a diameter of less than 10 mm, intended to be used as an imaging tool, may be rapidly eliminated by the kidneys [37], or a nanoparticle with a diameter greater than 200 nm may activate the complement system and be quickly removed from the blood, accumulating in the liver and spleen [33]. An in vivo biodistribution study showed that polymeric nanoparticles (RhB-CMCNP and RhB-CHNP), with slight negative charges and a size of 150 nm, tend to accumulate in tumor tissue more effectively [40].

The cytotoxicity of nanoparticles with different sizes and shapes, produced using the polymerization-induced self-assembly (PISA) methodology, was studied in nude mice bearing HT1080 human fibrosarcoma cells tumor xenografts.It was found that small spherical polystyrene core nanoparticles with a polyethylene glycol (PEG) corona (diameter of 21 nm) have higher tumor accumulation than the larger spherical nanoparticles (diameter of 33 nm) or their rodlike (diameter of 37 nm, contour length of 350–500 nm) or wormlike (diameter of 45 nm, contour length of 1–2 μm) counterparts [41].

Based on previous studies, the well-accepted size for long-term blood circulation is around 100 nm due to the size-dependent balance between organ filtration and tumor vessel extravasation [42]. This allows for prolonged circulation lifetimes of the nanoparticles, as well as their better accumulation in tumor tissue through the EPR mechanism, by reducing nonspecific interactions with blood components and other off-targeted cells and urinary excretion. The pores in tumor endothelium are much larger (100 nm to 1 μm, depending on the tumor type)than in the tight endothelial junctions of normal vessels (5–10 nm size), which results in enhanced permeability for relatively larger molecules (in the size range of 20–500 nm) passing through the vascular walls into the interstitium surrounding tumor cells [43].

Size-switchable nanoparticles were recently introduced to reduce their extravasation through the leaky tumor vasculature. The size change of such nanoparticles is triggered by internal (e.g., low pH in the tumor microenvironment oroverexpressed matrix metalloproteinases [MMPs]) or external (e.g., light, near-infrared [NIR] lasers, temperature, ultrasounds, magnetic fields) stressors and allowsfor prolonged blood circulation, nanocarrier storage at tumor sites, size reduction for penetrating tumor parenchyma, escaping from endo/lysosomes, and swelling or disassembly for drug release [42].

To achieve good efficacy with less toxicity, the “ideal” size for a nanoparticle as a drug carrier or a diagnostic tool should be determined individually while taking into account its cellular intake in the organism, distribution in the organism, and elimination from the organism.

### 2.2. Shape of Nanoparticles

Along with the size of the nanoparticles, their shape is also considered a major factor in their functionality [44]. Nanoparticles could be synthesized in various shapes, such as tubes, fibers, and spheres. Studies report that the shape of nanoparticles plays an important role in cellular uptake, blood circulation, anti-tumor activity, and biodistribution [45,46].

There are several reasons for characterizing the shape dependence of bio-nano interactions. One of them is a difference in the curvature structure of different-shaped nanoparticles. For example, rod-shaped nanoparticles have a larger contact area for cell membrane receptors than spherical nanoparticles, which, in turn, can reduce the number of available receptor sites for binding [47]. Another reason is related to the membrane wrapping time, which is greater for the elongated particles—for instance, cylindrical ones. Accordingly, spherical nanoparticles can be internalized more easily and faster through endocytosis compared to others [48]. In addition, it has been discovered that spherical nanoparticles are relatively less toxic [49,50]. It is shown that, in addition to the size, the shape of iron oxide (Fe_2_O_3_) nanoparticles is a major factor that contributes to particle cytotoxicity. It has been revealed that the use of rod-shaped Fe_2_O_3_ nanoparticles results in a higher extent of necrosis and a higher degree of membrane damage and reactive oxygen species (ROS) production.

It is reported that spherical nanoparticles demonstrate the highest uptake among these differently shaped nanoparticles [51]. Previous work has compared nanoparticles with identical surface area, ligand-receptor interaction strength, and grafting density of the polyethylene glycol and has found that spherical nanoparticles exhibit the fastest internalization rate, followed by cubic, and then rod- and disk-like, nanoparticles.

### 2.3. Surface Modification of Nanoparticles

Nanoparticle surface properties also have significant effects on their interactions with cellular compartments. Among these properties, nanoparticles’ surface charge and density could affect the degree of cellular uptake and particles’ interaction with biomolecules, as well [52,53]. Negatively charged structures bind less efficiently to cell surfaces than neutral or positively charged ones and, thus, they are characterized by smaller rates of endocytic uptake [54]. This is because electrostatic repulsive forces among the anionic surface and cellular environment repel the nanoparticles from entering. Moreover, nanoparticles’ surface charge also influences their toxicity properties in biological environments. It has been discovered that positively charged nanoparticles show more harmful effects in cell lines than negatively charged particles of a similar shape and size [55,56].

The hydrophilic/hydrophobic properties of nanoparticles are other key factors that play important roles in their ability to interact with biomolecules and cells. Several studies reveal that the surface membrane uptake of hydrophobic nanoparticles is favored [57,58] because the hydrophobic surface tends to interact with lipid tails (see Figure 3).

Surface chemical modification is necessary to provide the nanoparticles with a sufficient degree of stability in the physiological medium [59]. However, due to the high chemical reactivity of their surface, nanoparticles can influence cellular reactions in living systems. Moreover, surface functionalization is also necessary to prevent nanoparticles from aggregating [60] and reduce non-specific cell uptake [61]. In addition, the chemical modification of the surface is important for decreasing the toxic effects of nanoparticles and achieving a safe-design strategy that plays a crucial role in their potential applications [62,63].

### 2.4. Protein Corona

Another important feature of the biological environment is that the nanoparticles, while exposed to biological fluids, such as blood, lymph, gastric juice, etc., are covered with a kind of “corona”—a layer of proteins that are constantly adsorbed on their surface [64]. Undoubtedly, this protein (biological macromolecule) can mask several physicochemical properties (such as the surface charge and solubility) of nanoparticles and make them recognizable to the immune system, which may neutralize them before they reach the target site [65]. The covering of nanoparticles with these proteins largely determines their biological identities and further fate—their distribution through tissues and organs, their rate of excretion from the body, and their opsonization (phagocytosis involving membrane receptors). This is the key factor causing differences in the performance of nanoparticles within in vitro and in vivo [66] studies, where nanoparticles indicate good results in the first case and usually are worse or non-effective in living organisms [67,68].

However, the composition, density, and mechanism of formation of this macromolecular layer depends not only on the physicochemical characteristics of the nanoparticles, association/dissociation constants, competitive binding processes, and species of proteins in the biological medium but also on where the nanoparticle is located [69,70].

The formation of this protein corona depends not only on the composition of the nanoparticle—its size, shape, surface state, and exposure time—but also on the type of media, the nanoparticle–to–protein ratio, and the presence of ions and other molecular species that interfere in the interaction between proteins and nanoparticles [71].

Careful modulation of the protein corona through the tuning of intrinsic and extrinsic parameters may allow nanoparticles to be delivered optimally in tumor tissue [72]. A study [73] has shown that linking Z domain-based recombinant affibody (RA) scaffolds to magnetosomes (BMPs), which were then preadsorbed on anti-HER2 humanized mAb (TZ) and incubated with human plasma, resulted in the more effective targeting capability and drug delivery of nanoparticles in a simulated in vivo environment by incorporating the majority of TZ molecules into the nanoparticle–corona complex. It is also crucial to consider external factors, such as temperature, pH, and shear stress, for developing a more efficient protein corona [74]. Studies on protein corona composition on nanoparticles suggest that choosing the right materials can influence the types of proteins adsorbed from plasma and improve delivery effectiveness [72].

In a dynamic process, protein corona is usually categorized into hard (irreversible) and soft (reversible) forms [67], depending on their affinity properties and periods of protein exchanges (see Figure 4).

Proteins in the hard corona are characterized by high affinity on the surface of the nanoparticles and longer exchange times. In contrast, the soft corona represents weakly connected proteins with low affinity on the nanoparticle surface and short exchange periods [67].

The nanoparticle–protein layer is a formation that “cells see” [75]; moreover, it plays a significant role in determining the interactions of the nanostructures with the biological environment. In the majority of the nanoparticles, cellular uptake is correlated with the hard corona properties, and they could develop the interactions occurring among nanoparticles and proteins, phospholipids, membranes, and DNA [76]. In other words, the corona can “control” the process of choosing the biomolecule types and interaction forms. Moreover, the nanoparticle–protein hybrid structure and cell receptor interactions primarily depend on the corona. Some of these interactions emerge due to the presence of nanoparticles and are not inherent to natural biosystems. Therefore, protein coronas that form on the surface of the nanoparticles could stimulate or mitigate an immunological response [77], which is a crucial aspect of understanding and evaluating their cytotoxic effects [78]. Consequently, understanding the formation of the protein corona and controlling its properties and interactions with nanoparticle surfaces and other proteins are crucial factors for the application of nanostructures in biomedicine [79,80].

## 3. Conclusions

The current knowledge of nanoparticles as drug carriers or diagnostic tools shows that generalized optimal physicochemical features for more potent nanoparticles cannot be recommended. However, practical design principles based on this knowledge can be identified, which can guide the achievementof reduced premature clearance of nanoparticles from blood circulation, less systemic toxicity, and high tumor penetration.

It is well established that small-sized nanoparticles are easily eliminated from blood circulation through renal filtration without reaching the desired concentration at the target site. Therefore, the consensus is that nanoparticles must be larger than 10 mm. Conversely, it is also well established that as the particle size increases, nanoparticle uptake by the reticuloendothelial system (RES) increases—nanoparticles with a diameter greater than 200 nm may activate the complementary system and be quickly removed from blood circulation, accumulating in the liver and spleen. Therefore, the optimal size of a nanoparticle should be determined on an individual basis while taking into account important factors, such as the amount and rate of cellular uptake, intracellular trafficking, cytotoxicity, tumor penetration, blood circulation half-time, the disease site, and the expected therapeutic goals. This condition is true for the optimal shape and surface modification of nanoparticles.

Further research is required to find optimal combinations of the different physicochemical properties of various nanomaterials in physiological settings, taking into account the target sites in an organism and the goals imposed on them. The design strategy for developing a potent nanoparticle might need to consider the opposing requirements for its physicochemical properties (depending on the intercellular space) where the tunability properties of nanocarriers are advantageous, as their surfaces can be functionalized appropriately to fulfill size requirements. The development of size-switchable intelligent nanoparticles that are activated by the intrinsic tumor micro-environment offers challenging prospects. The potential of endogenous size-tuning triggers (such as curtain enzymes, glucose concentrations, redox reactions, temperatures, and pH differences) should be further explored, whereas some external stressors, such as temperature and light, can damage normal cells and organs.

A better understanding of the importance of the interaction of nanoparticles with the biological environment, after entering blood circulation and targeting tumor tissue, will play a decisive role in designing highly potent nanoparticles. The properties of the nanoparticle surface are one of the main parameters that affect their interaction with cells and body fluids.

The surface load and its density should also be the focus of the design process, as it can affect the degree of the absorption of nanoparticles by the cell, as well as their interaction with biomolecules. The attachment frequency of negatively charged structures with the cell surface is lower than that of neutral or positively charged particles. The neutral nanoparticles are coated with an immune protein corona that retains more IgG or fibrinogen than charged nanoparticles, and an elevated interaction with the immune system is achieved. Another interesting point is that negatively charged nanoparticles are removed from the body faster than positively charged particles, which leads to elevated cytotoxicity.

Another important parameter is the chemical modification of the nanoparticle surface, which allows for a reduction in the toxicity of nanoparticles used for biomedical purposes by controlling and modulating cell penetration and reducing the aggregation of nanoparticles in biological fluids. Further, the presence of large amounts of lipids, sugars, nucleic acids, and especially proteins in biomolecules can significantly affect the stability of nanoparticles in the physiological environment; surface modification plays an important role in solving this problem.

## Figures and Tables

**Figure 1 jfb-13-00196-f001:**
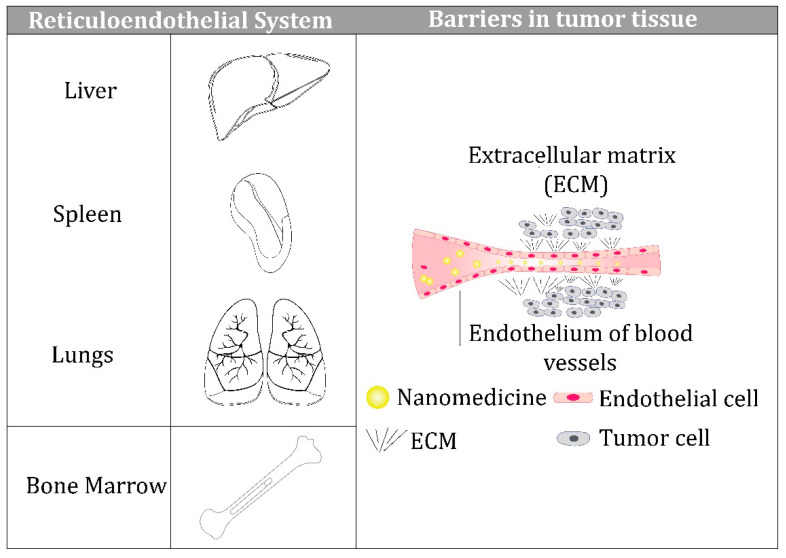
The barriers that prevent intravenously administered nanoparticles from reaching their target include the reticuloendothelial system (RES), comprised of the liver, spleen, lungs, and bone marrow; the endothelium of blood vessels within the target tissues; and the extracellular matrix (ECM) of the target tissue.

**Figure 2 jfb-13-00196-f002:**
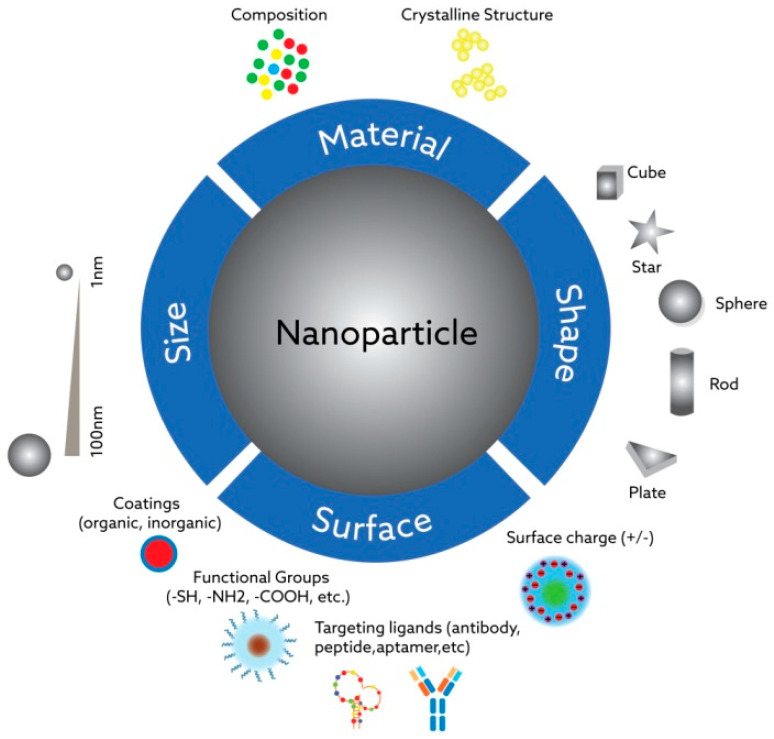
The basic features of nanoparticles support their interactions with biological systems: The interactions between nanoparticles and biological systems are influenced by particle size and shape, coating material properties, morphology, surface load and the presence of functional groups, hydrophilic/hydrophobic properties, chemical composition, and crystalline structure.

**Figure 3 jfb-13-00196-f003:**
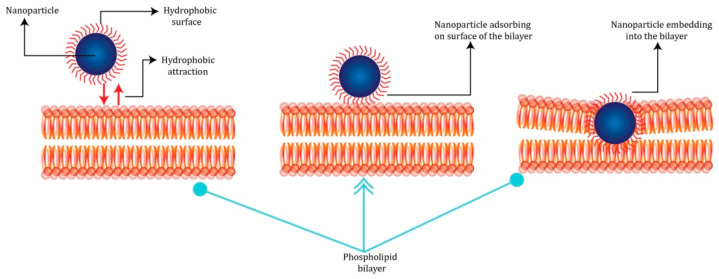
Process of inclusion of nanoparticles with the hydrophobic surface into the membrane. A hydrophobic contact is formed between the hydrophobic layer and the phospholipid layer on the surface of the nanoparticle, which is due to the absorption of the nanoparticle by the cell membrane. This process, in turn, causes the deformation of lipid bilayers and the translocation of the nanoparticle into it.

**Figure 4 jfb-13-00196-f004:**
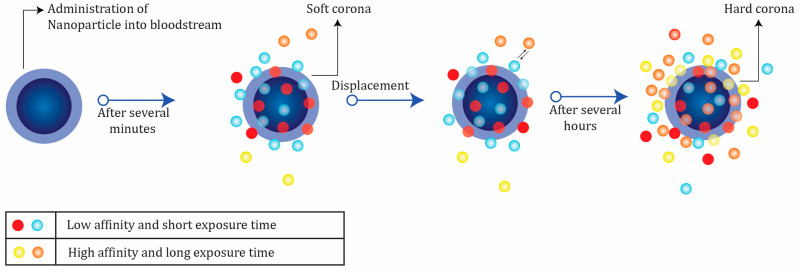
The formation of protein corona on the nanoparticle surface.Immediately after the in vivo administration of nanoparticles, low-affinity proteins in high concentrations are absorbed on their surface;these are replaced by proteins with a higher affinity after a long exposure time (the Vroman effect). Low-affinity proteins form a so-called “soft” corona layer, while the high-affinity proteins form a strongly bound layer of “hard” corona. Hard corona proteins are located closest to the nanoparticle surface, which makes them more sensitive to thermodynamic changes (irreversible) depending on the functionality, hydrophobicity, or hydrophilicity of the nanoparticle and the nature and temperature of the biological fluid. The soft corona is not directly connected to the surface of the nanoparticle, and only weak interaction forces are present.

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
