# Peer review of "Overview of Physicochemical Properties of Nanoparticles as Drug Carriers for Targeted Cancer Therapy"

_jfb, 2022, doi:10.3390/jfb13040196_

Round 1

Reviewer 1 Report

A nice review of the important and often overlooked topic of factors determining the toxicity of nanoparticles in biological systems. The manuscript is well-organized and presents the current knowledge in a clear manner.

Nanoparticles (and polymers) differ fundamentally from traditional small molecules by being in a size range, where the size and shape alone can cause toxicity. This is counterintuitive to the normal ideas of toxicity, where reactivity due to chemical factors are the cause. The problem arises because the immune system rely on a number of different triggers, where one of them is "could this be a foreign biological object?" and this response is often related to size and shape - if something has a size and shape similar to bacteria or vira, then the immune system reacts and this reaction can be dangerous. Adhesion of proteins causing a "corona" can also add to the problems.

There are not that many reviews on these topics for non-specialists and this is a nice review of the important and often overlooked topic of factors determining the toxicity of nanoparticles in biological systems.

Reviewer 2 Report

In their manuscript dr. Yagublu et al. reviewed some physico-chemical properties of the nanoparticles that might play decisive roles in their interaction with the biological systems, like size, shape, surface properties (charge, density), nanoparticle-protein layer. The review is clear, comprehensive and of relevance to the field. There were not many similar reviews published recently and, this current review is relevant and of interest to the scientific community. The iconography (figures/tables/images/schemes) is appropriate, and easy to interpret and understand. Statements and conclusions are drawn coherent and supported by the listed citations. The cited references are relevant, the most published within the last 5 years: the bibliography does not include many self-citations. However, for a better understanding of the manuscript, the authors are kindly asked to include some information for describing, in few words, the cell lines used in the reviewed studies (HT1080,  MC3T3-E1 or PC12 cell lines), like species, histological type.

Author Response

Dear Reviewer, thank you for your valuable comments. The quality of our manuscript has been improved through changes and additions.

Reviewer 3 Report

Despite the manuscript is an overview of the main properties of nanoparticles and their effect in drug delivery performance, I consider that the information provided by the authors is missing some relevant literature and recent findings on the area. In general, shape and size effect on the biological interactions of nanoparticles have been widely described in several review works since the last decade. From the title of the manuscript, as a reader I expected to find some recent advances on how physical and chemical properties of NP affect their performance as drug carriers in cancer treatment, nevertheless the works is strongly focused on the nano-bio interactions in general. Some important points such as zeta potential of the surface, roughness, and differences between organic an inorganic NP were left out of the manuscript, and I suggest the authors to consider their inclusion. Here are my specific comments.

1.       References 4 and 5 can be omitted, since it is well known the two areas where NP can be used, besides ref. 5 is about gold NP only.

2.       Figure 1 could be improved, I could be more useful to use a schematic representation, rather than the figure of human body, the information in the small box is too small for reading.

3.       The way the information appears in lines 73-76 should be improved. The references, as presented there, seems to be citing to what RES and ECM are, but actually references 19-23 are specific works on how NP interact with those biological barriers. If the intention of the authors here is to provide references about how RES and ECM are formed and located, they could cite some work related to the basic concepts on those structures.

4.       The introduction section should provide a concise idea of what the review will be about, the last paragraph mentions some isolated ideas, where for instance protein corona is not considered.

5.       It is hard to understand the idea of fig.2, what is the meaning of the circles? Are the circles representing NP? what is the meaning of the yellow branches? And what about the arrows? Is there any reason for drawing the arrows from left to right? since the authors are including these statement in the caption “These features of nanoparticles severely affect endocytosis, distribution, selectivity, and toxicity in the bio environment.”, then this processes could be included in the diagram/scheme.

6.       Line 122: Is the sentence “where the as the cell membrane is mostly permeable for small non-polar molecules” correct?

7.       Lines 128-145, I suggest not to mix the reference style (Author-year).

8.       Lines 175-178. There are studies on this matter (effect of shape) dating since 2010, so I believe that “recent studies report…”makes no sense here. This is something that has been studied long ago (for example https://doi.org/10.1016/j.micromeso.2012.05.040, https://doi.org/10.1016/j.biomaterials.2009.09.060) and there are more papers that are relevant on this topic, since Ref. 47 is about one specific type of nanoparticle, please improve the literature review on this feature and talk about those challenges that remain unclear so far. What about triangle or cubic-shaped nanoparticles?

9.       Line 188, use subscripts in (Fe2O3)

10.   With regard to section 2.4, I suggest providing more information on what have been done on how protein corona interferes with the delivery of drugs for cancer treatment, since this topic is briefly approached. Here are some suggestions of other works that summarize some relevant findings. https://doi.org/10.1016/j.jconrel.2022.03.056 , https://doi.org/10.1039/D0RA05241H; https://doi.org/10.2147/IJN.S273721  

Author Response

Dear Reviewer, thank you for your valuable comments. We deeply appreciate your support of our study and your assistance in revising it. Your feedback was acknowledged one by one, and it greatly improved the standard of the work that was submitted.

Round 2

Reviewer 3 Report

After reading the revised version of the manuscript, I consider that the authors have addressed some points that were left out in the previous version. The manuscript has been improved.